# 60 million years of glaciation in the Transantarctic Mountains

**Iestyn D. Barr** [1,2] ✉, **Matteo Spagnolo** [3], **Brice R. Rea** [3], **Robert G. Bingham** [4], **Rachel P. Oien**[3], **Kathryn Adamson**[1,2], **Jeremy C. Ely** [5], **Donal J. Mullan**[6], **Ramón Pellitero** [7] & **Matt D. Tomkins**[8,2]

The Antarctic continent reached its current polar location ~83 Ma and became shrouded by ice sheets ~34 Ma, coincident with dramatic global cooling at the Eocene-Oligocene boundary. However, it is not known whether the first Antarctic glaciers formed immediately prior to this or were present significantly earlier. Here we show that mountain glaciers were likely present in the Transantarctic Mountains during the Late Palaeocene (~60–56 Ma) and middle Eocene (~48–40 Ma). Temperate (warm-based) glaciers were prevalent during the Late Eocene (~40–34 Ma) and, in reduced numbers, during the Oligocene (~34–23 Ma), before larger, likely cold-based, ice masses (including ice sheets) dominated. Some temperate mountain glaciers were present during the Miocene Climatic Optimum (~15 Ma), before a widespread switch to cold-based glaciation. Our findings highlight the longevity of glaciation in Antarctica and suggest that glaciers were present even during the Early-Cenozoic greenhouse world.

During the Late Cretaceous (~100.5–66 Ma), the global climate was considerably warmer than present[1] and allowed dense sub-tropical forests to occupy Antarctica, despite the continent's polar location[2]. During the Cenozoic (past ~66 Ma), a gradual reduction in atmospheric $CO_2$[3,4] pushed Earth from a greenhouse to icehouse climate[5,6]. The most dramatic period of cooling occurred at the Eocene–Oligocene (EO) boundary (~34 Ma) and led to the growth of ice sheets in Antarctica[3,7]. Given the key role of Antarctic ice sheets in shaping global climate[8], and controlling global sea level (SL), their development and evolution during and after the EO-boundary has been the focus of considerable research[3,7,9]. However, far less is known about the evolution of early mountain glaciers in Antarctica, despite them being the first ice masses to occupy the continent, acting as inception points for ice-sheet growth[10–12], and serving as key palaeoclimatic indicators. A better understanding of these glaciers can help to address some fundamental and unanswered questions about Antarctica's glacial history[13], including: when was the onset of glaciation? Were glaciers present during full greenhouse conditions? For how long did temperate (i.e. warm-based) mountain glaciers exist?

Extensive ice-sheet cover and widespread erosion have often hindered the study of former mountain glaciers in Antarctica by obscuring and eroding glacial deposits[14]. As a result, many previous efforts to understand the continent's long-term glacial history are based on distal proxies, including offshore sediment records[15], marine isotope data[16], and indicators of eustatic SL fluctuations[13]. However, these records are unable to fully capture information about smaller ice masses such as those that formed at the onset of glaciation. Here, we show that glacial cirques (armchair-shaped, glacially-eroded depressions that form at the onset of glaciations), which are ubiquitous in Antarctica[11,17,18], can help to fill this knowledge gap[10,11], revealing information about the early glacial history—including the transition from greenhouse to icehouse conditions, a period for which there are few high-latitude terrestrial records[19]. In particular, cirque altitudes can provide information about glacier equilibrium line altitudes (ELAs)

[1]Department of Natural Sciences, Manchester Metropolitan University, Manchester, UK. [2]Cryosphere Research at Manchester, Manchester, UK. [3]School of Geosciences, University of Aberdeen, Aberdeen, UK. [4]School of GeoSciences, University of Edinburgh, Drummond Street, Edinburgh, UK. [5]Department of Geography, University of Sheffield, Sheffield, UK. [6]School of Natural and Built Environment, Queen's University Belfast, Belfast, Northern Ireland, UK. [7]Departamento de Geografía, Universidad Nacional de Educación a Distancia (UNED), Madrid, Spain. [8]Department of Geography, University of Manchester, Manchester, UK. ✉e-mail: i.barr@mmu.ac.uk

during former periods of small-scale (cirque-confined) mountain glaciation[11,20], which can, in turn, be used to obtain quantitative information about past climates[21].

In this study, we analyse the altitudes of glacier-free cirques across the Transantarctic Mountains (TAM) and use this information to estimate the associated glacier ELAs. Assuming that these cirques developed when occupied by temperate glaciers (see the "Methods" section)[20] and using modern temperate glaciers (and the climatic conditions they experience) as analogues[11], we reconstruct palaeotemperatures from these ELAs. Through comparison with published palaeotemperature data, we then establish when glaciers first formed in the TAM and estimate where and when subsequent temperate (i.e. warm-based) mountain glaciers were present (note: henceforth, the term 'mountain glacier' is used to imply a small, cirque-confined glacier).

We show that mountain glaciers were likely present in the TAM during the Late Palaeocene (~60–56 Ma) and middle Eocene (~48–40 Ma). Temperate (warm-based) glaciers were widespread during the Late Eocene (~40–34 Ma) and, in reduced numbers, during the Oligocene (~34–23 Ma). By the Early Miocene (~23–15 Ma) it is likely that all of the cirques in the TAM were submerged by ice sheets, and/or occupied by mostly cold-based glaciers, but that some temperate glaciers were present during the Miocene Climatic Optimum (~15 Ma),

before a widespread switch to cold-based glaciation, which continues to the present day.

## Results and discussion
### Climate during former periods of mountain glaciation

In total, 14,060 glacial cirques were identified and mapped across the TAM. Of these, 1292 were classified as glacier-free, with past mountain glacier ELAs ranging from 107 to 4173 m in the modern topography. These were then modified to reconstructed palaeo topography (see the "Methods" section), with resulting ELAs ranging from 211 to 3888 m (Supplementary Fig. 1). Assuming that the mean summertime air temperature (MSAT) at each modified ELA during the former occupation by temperate mountain glaciers was 3.6 ± 2.5 °C (see the "Methods" section) implies MSAT increases (relative to present) of between 10.0 ± 2.5 and 40.8 ± 2.5 °C, depending on cirque location and altitude (Supplementary Fig. 1). When corrected to SL, palaeo MSATs range from 5.0 ± 2.5 to 28.9 ± 2.5 °C (Fig. 1a). Our results indicate that the presence of temperate mountain glaciers in the TAM required notably increased MSAT, relative to the present. However, as with a number of other cirque populations globally[20], the range of SL MSATs (Fig. 1a) implies that temperate mountain glaciation occurred at different times in different places—e.g. higher altitude cirques were likely occupied by temperate mountain glaciers earliest during the onset of

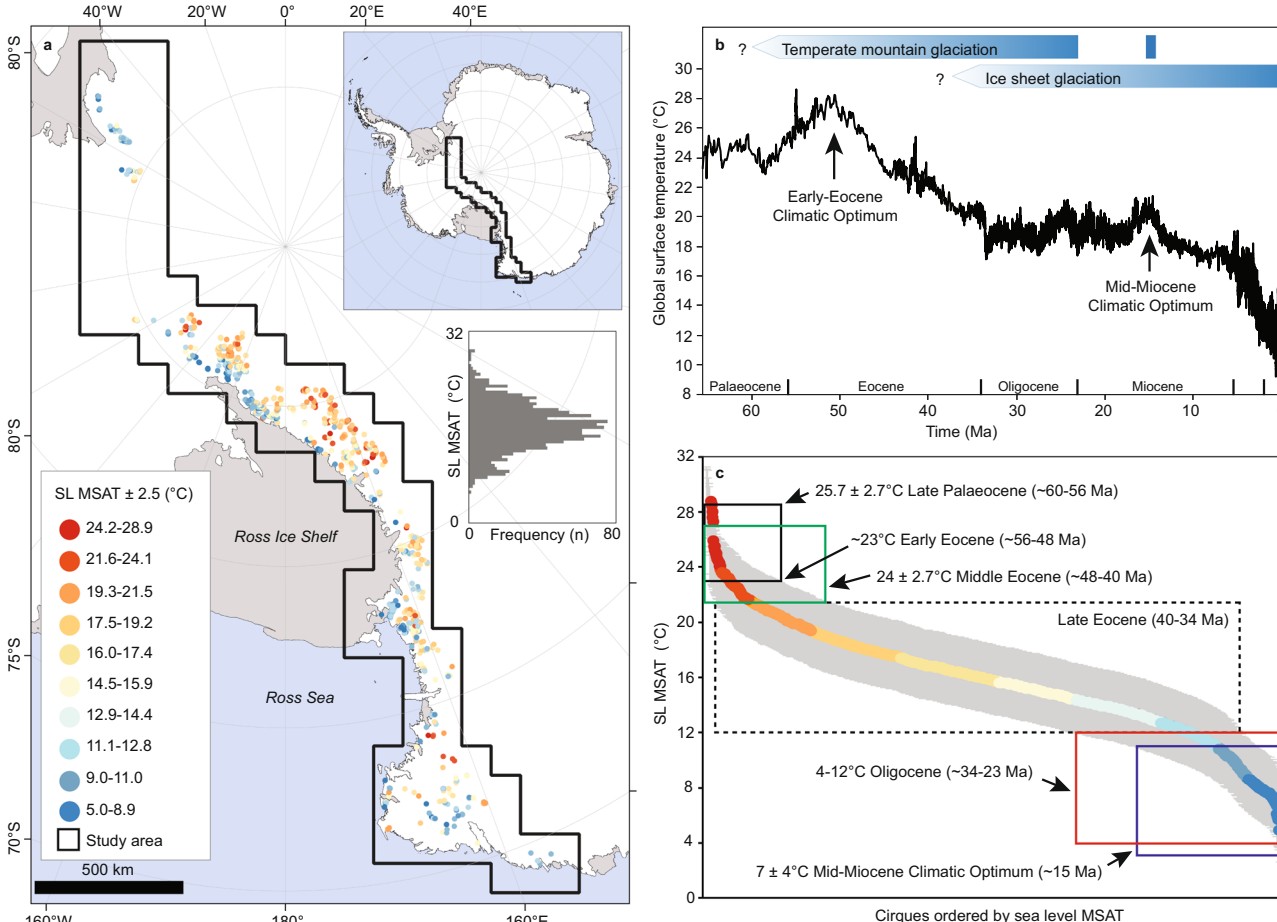

**Fig. 1 | Palaeotemperature reconstructions from Antarctica and globally.**
**a** Glacier-free cirques (n = 1292) in the Transantarctic Mountains coloured according to the sea level mean summer air temperature (SL MSAT) required for them to be occupied by temperate (warm-based) mountain glaciers. The inset histogram shows the frequency distribution of SL MSAT. Antarctic coastline data from the SCAR Antarctic Digital Database, accessed [2021][49] license: https://creativecommons.org/licenses/by/4.0/. **b** Global surface

temperature data for the Cenozoic[4], highlighting key climatic periods referred to in the text, and our interpretation of the glacial history of the TAM.
**c** Cirques ordered by SL MSAT (grey area represents the ±2.5 °C MSAT uncertainty) required for them to be occupied by temperate mountain glaciers. Colours are based on the values in subplot (**a**). Antarctic temperature estimates for different time periods in subplot (**c**) are based on published data from biological proxies (Supplementary Table 1).

glaciation (when MSATs were higher), followed by sequentially lower altitude cirques. Through comparison with published estimates of SL palaeo MSAT derived from biological proxies (Supplementary Table 1), we use our ELA-based SL MSAT estimates to constrain the timing of mountain glaciation across the TAM since the Late Cretaceous. During this period, the Antarctic climate not only progressively cooled (with occasional fluctuations) but also experienced increased aridity[19]. However, for much of the time, mean annual precipitation remained within limits characteristic of those that sustain present-day temperate glaciers[21].

### The onset of glaciation

At the start of the Cenozoic, during the Palaeocene (~66–56 Ma), the Antarctic and global climate experienced warming towards the Early-Eocene Climatic Optimum (53–50 Ma, Fig. 1b). By the Late Palaeocene (~60–56 Ma), fossil plant evidence suggests Antarctic warm month mean temperatures (taken here conservatively as MSAT) of 25.7 ± 2.7 °C[22] (Supplementary Table 1). Our study indicates that such temperatures were not conducive to widespread glaciation in the TAM, though it is likely that some high-altitude cirques (total = 41, i.e. ~3%) were occupied by temperate mountain glaciers, supported by comparatively high mean annual precipitation (~2110 mm[22], Supplementary Table 2), and a very small group (total = 3, i.e. <1%) were potentially occupied by cold-based glaciers (Figs. 1c and 2a). Thus, the onset of glaciation in the TAM most likely began in the Late Palaeocene. During the Early Eocene (~56–48 Ma), fossil plants reveal Antarctic warm month mean temperatures of ~23 °C[23], suggesting similar climatic and glacial conditions to the Late Palaeocene (Figs. 1c and 2a). By the Middle Eocene (~48–40 Ma), warm month average temperatures of 24 ± 2.7 °C[22] will have facilitated more widespread temperate mountain glaciation (total cirques = 105, i.e. ~8%, Figs. 1c and 2b), but this may have been partly countered by a reduction in mean annual precipitation (to ~1534 mm[22], Supplementary Table 2). Climate cooling through the Late Eocene (~40–34 Ma[22]) is likely to have triggered a period where the majority of cirques in the TAM were occupied by temperate mountain glaciers (total = 985, i.e. ~76%). Some may have remained glacier-free, while those at higher elevations were likely engulfed by larger (possibly cold-based) ice masses (Figs. 1c and 2c). Again, the extent of glaciation during this period may have been partly restricted by a continued reduction in mean annual precipitation (to ~1000 mm[19], Supplementary Table 2). However, there is strong geological evidence for glaciation of different parts of Antarctica during the Late Eocene[24], including ice-rafted debris from the South Orkney microcontinent. This indicates marine-terminating glaciers along the Weddell Sea Embayment as early as 36.6 Ma[15], supporting our view that temperate mountain glaciers were likely ubiquitous in the TAM. In addition, thermochronologic studies find evidence of enhanced exhumation in the TAM during the Late Eocene and Early Oligocene (see below), presumed to reflect increased erosion by temperate glaciers[25,26].

### Shift to ice-sheet glaciation

At the EO-boundary (~34 Ma), global climate[4] and the climate of Antarctica[19] shifted towards notably colder and drier conditions, and there is abundant evidence that glaciers coalesced to form ice sheets[3,7]. Biological proxies suggest Antarctic MSATs of 4–12 °C[27–29] and mean annual precipitation of 500–800 mm[19] during the Oligocene (~34–23 Ma) (Supplementary Tables 1 and 2). The analyses presented here suggest that these MSATs would have resulted in all TAM cirques being occupied by glacial ice, though only a minority (n = 196, i.e. ~15%) were likely occupied by temperate mountain glaciers, while all others would have been occupied by cold-based ice (Figs. 1c and 2d). Given limited mean annual precipitation, these glaciers likely experienced comparatively low levels of accumulation, but this was still sufficient to support glaciation since some present-day temperate mountain

glaciers exist under comparable conditions[21]. Throughout the Oligocene and into the Early Miocene, the climate of Antarctica experienced cooling and increased aridity[19]. By the Early Miocene (~23–15 Ma) it is likely that all of the cirques in the TAM were submerged by ice sheets, and/or occupied by mostly cold-based glaciers. Thus, temperate mountain glaciation in the TAM effectively ceased.

### The return of temperate mountain glaciers

During the Mid-Miocene Climatic Optimum (~15 Ma), the Antarctic climate switched from its cryo-arid state and experienced a notable period of warming and reduced aridity[4]. Fossil wood from the TAM, and pollen records from the Ross Sea, imply summer temperatures of 7 ± 4 °C[30,31], which coincided with mean annual precipitation of ~600 mm[19]. Results from the present study suggest that these conditions may have triggered the return of temperate mountain glaciers to some cirques in the TAM (n = 141, i.e. ~11%) (Fig. 2e). This assertion is supported by glacial and fossil evidence from the Dry Valleys (part of the TAM) which suggests that a smaller East Antarctic Ice Sheet coincided with warm-based mountain glaciers during the Middle Miocene[32]. However, by ~13.96 Ma, a reduction in temperatures and increased aridity (e.g. MSAT of −1.7 °C and mean annual precipitation of 150 mm in the Dry Valleys) caused a switch in glacier thermal regime from warm- to cold-based[9,19,32]. This was likely the termination of temperate mountain glaciation in the TAM, as the climate progressively cooled and dried through the Pliocene (~5.3–2.6 Ma) and Pleistocene (~2.6 Ma–11.7 ka)[33]. Some of the (now) glacier-free cirques investigated in this study may have been occupied by cold-based ice for much of this time, and only exposed subaerially due to Plio-Pleistocene bedrock uplift[9] and/or comparatively recent (i.e. Holocene, ~11.7 ka to present) ice-sheet thinning[34]. Following subaerial exposure, the region's extreme aridity prevented the formation of glaciers within these cirques[9], and they will likely remain devoid of glacial ice until the future climate is conducive to temperate mountain glaciation.

Overall, our findings help to address long-standing uncertainty about the timing of glacier initiation in Antarctica and demonstrate that mountain glaciers were likely present at high elevations during the Early Cenozoic when sub-tropical vegetation occupied much of the continent[3], and that warm-based mountain glaciers were widespread in the TAM even prior to the EO-boundary. As such, we provide insight into Antarctica's glacial history and support previous studies which indicate early glaciation of the continent[6,13].

## Methods

### Cirque identification and mapping

Cirques were mapped throughout the TAM using the Reference Elevation Model of Antarctica (REMA) Digital Surface Model (DSM), which has an 8 m spatial resolution and a vertical error of <1 m[35]. Cirques were identified as large depressions, bounded upslope by arcuate headwalls, and open in a down-valley direction[36]. Initially, all features classifiable as cirques in the REMA DSM were mapped, irrespective of the degree of glacier cover. For all cirques, headwalls (i.e. their arcuate upper margins) were identified and mapped following established procedures[20]. For cirques with minimal glacier cover, the cirque threshold (i.e. the break-of-slope which often marks a cirque's lower limit) was also mapped, again following established procedures[20]. Where thresholds were not identifiable, the lower extent of each cirque was drawn as a straight line connecting the outer limits of the headwall or lateral spurs (examples of mapped cirques are shown in Supplementary Fig. 2, and all cirques are presented as Supplementary Data). Based on the above approach, we compiled the first comprehensive database of cirques across the entirety of the TAM. Published cirque maps exist for a small number of features (<60 cirques), only in sub-regions, and were used for cross-validation where possible[17,18].

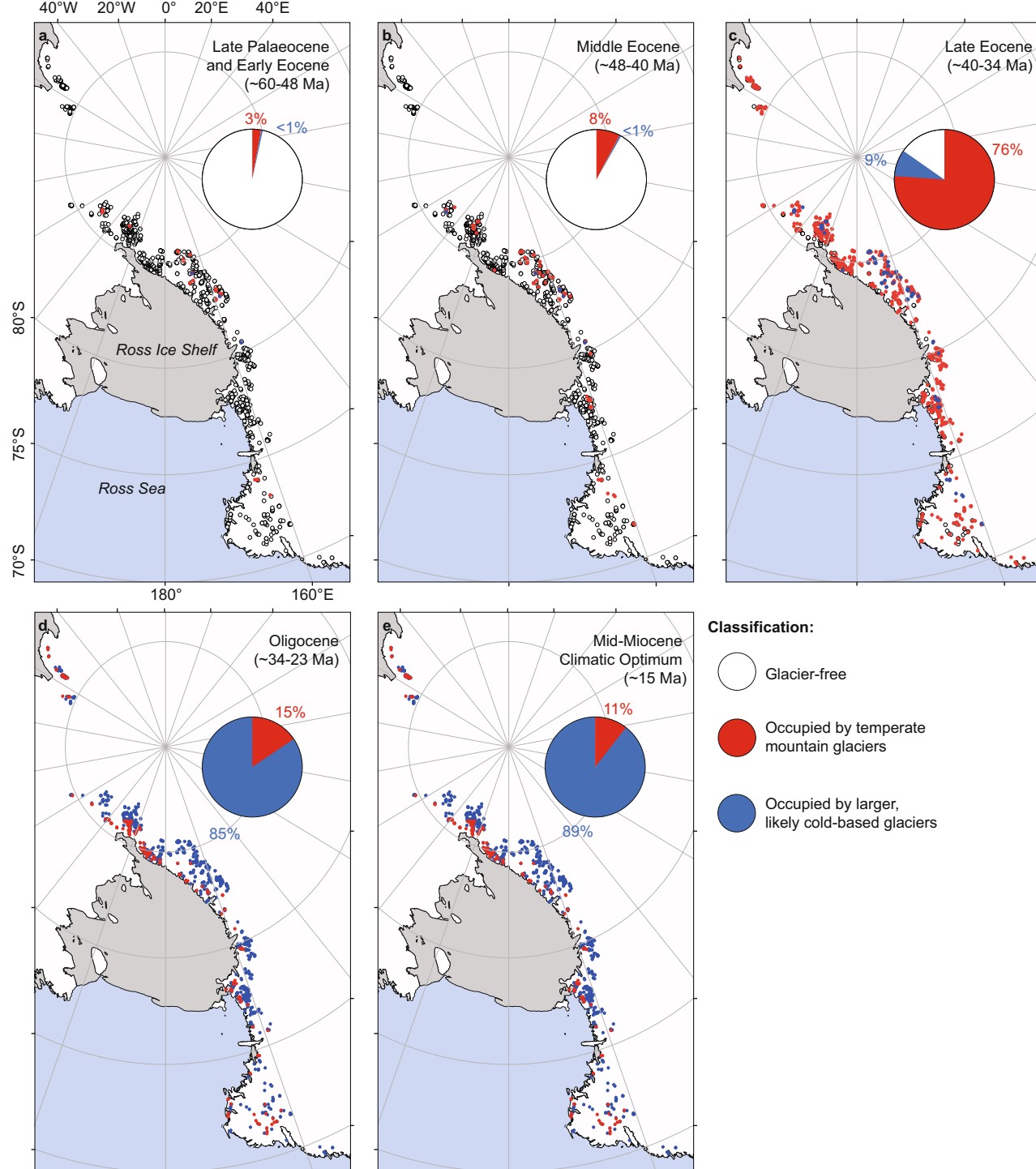

**Fig. 2 | Glacier-free cirques in the Transantarctic Mountains are classified according to their glacial status during various periods of the Cenozoic. a–e** Pie charts show the percentage of cirques classified as glacier free (white), occupied by temperate mountain glaciers (red %), and occupied by larger, likely cold-based glaciers (blue %) for the total population (*n* = 1292). Antarctic coastline data from the SCAR Antarctic Digital Database, accessed [2021][49] license: https://creativecommons.org/licenses/by/4.0/.

## Estimating glacier cover

To investigate the early evolution of Antarctic mountain glaciers, we focused only on glacier-free cirques. These landforms provide a unique record of past temperate mountain glaciation (including the sculpting of the initial glacier-free landscape) and are unlikely to have been modified extensively during subsequent glacial-interglacial periods[37], in part due to low rates of sub-aerial weathering during hyper-arid ice-free conditions[38], and because they, and much of the

Alpine topography of the TAM, were often occupied by minimally erosive cold-based ice[39,40]. These characteristics also apply to glacier-occupied cirques, but they are not considered here since the presence of ice of unknown thickness limits our ability to measure meaningful cirque altitudes. To assess the degree of glacier cover within each mapped cirque, we used a published rock outcrop dataset[41]. For most of the study region, this dataset was compiled through automated mapping from Landsat 8 satellite imagery (processed at 30 m spatial

resolution), with an overall pixel classification accuracy of 74 ± 9%[41]. At higher latitudes (i.e. above -83°S), we used a rock outcrop database compiled through manual identification and digitisation from topographic maps (available via http://www.add.scar.org[41]). For our purposes, all areas not classified as bedrock were classified as glacial ice. This conservative approach is likely to over-estimate glacier cover since areas covered by non-glacial ice (e.g. ice patches) and/or perennial snow will also be classified as glacial ice. Given this potential, we classified cirques as glacier free if <20% of their surface area was covered by glacial ice[37].

## Palaeo-glacier equilibrium line altitudes

For each glacier-free cirque, the single lowest and single highest DSM grid cells were recorded as the minimum and maximum altitudes, respectively[42]. From this, former mountain glacier ELAs were calculated using the toe-to-headwall altitude ratio (THAR) method. This approach assumes that a glacier's ELA lies at a certain proportion of the altitude between its minimum and maximum points[43]. Given that the present study relies on cirque altitudes, which are a good indicator of former mountain glacier minimum altitudes but are likely to overestimate their maximum altitudes, we adopt a comparatively low THAR of 0.35[43]. Since our interest is in Antarctica's long-term glacial history, each former ELA was modified using nine different reconstructions of palaeo topography (i.e. minimum, median and maximum reconstructions of topography at 34, 23, and 14 Ma)[44]. In the manuscript, we report results based on the median reconstruction of topography at 34 Ma given the focus on conditions prior to widespread ice-sheet development in Antarctica, and since the maximum and minimum reconstructions for this period represent extremes. In the Supplementary Information (Supplementary Figs. 3 and 4) we demonstrate the impacts on study findings of selecting each of the different topography reconstructions. Differences between ELAs in the present-day topography and ELAs modified to the median 34 Ma topography range from 1037 to 2161 m, with a mean of 392 ± 488 m (±1$\sigma$). Though the EO-boundary might not precisely reflect conditions during the initial period(s) of temperate mountain glaciation, this topography is more representative than the present-day, given ~34 million years of subsequent ice-sheet loading, volcanism, thermal subsidence, horizontal plate motion, erosion, sedimentation, and flexural isostatic adjustment, all of which are considered in the EO topographic reconstruction[44].

## Present-day temperatures

At each modified ELA, present-day MSAT (December, January, and February) spanning the 1981–2010 period was estimated from gridded (0.25° × 0.25°, 2 m above the surface elevation) ERA5 reanalysis data[45], using altitude-dependent present-day lapse rates for Antarctica (i.e. −4.1 °C km$^{-1}$ for altitudes of 0–1000 m; −6.2 °C km$^{-1}$ for 1000–2000 m; −8.9 °C km$^{-1}$ for 2000–3000 m; and −9.1 °C km$^{-1}$ for >3000 m)[46].

## Palaeotemperature reconstruction

Because of the requirement for efficient and focused subglacial erosion (driven by localised sliding and the flushing of subglacial debris), it is generally assumed that cirques primarily develop (i.e. initiate and grow) when they are occupied by comparatively small temperate (i.e. warm-based) mountain glaciers[20,37]. Thus, we presume that the climatic conditions which facilitate present-day temperate glaciation in various regions globally also prevailed in the TAM when the ice-free cirques analysed in this study were being formed[11]. Following this logic, we note that, based on a global dataset of examples[21], the MSAT at the ELA of present-day temperate glaciers (here defined as glaciers <60° latitude) is 3.6 ± 2.5 °C (Supplementary Table 3). This value represents the mean ± 1$\sigma$, but it is worth noting that the present-day MSAT values for these temperate glaciers (Supplementary Table 3) range from −3.7 to 8.8 °C.

Though this 12.4 °C range is considerable, the 10th–90th percentile range is only 5.9 °C (i.e. 0.9–6.8 °C) and is very similar to the mean ± 1$\sigma$ range. For this reason, and following previous studies of Antarctic cirques[11], we assume that this mean ± 1$\sigma$ MSAT also applied to the ELAs of former temperate mountain glaciers in Antarctica[11]. To facilitate comparison with previous palaeoclimate studies, these palaeo-MSATs were corrected to SL using a lapse rate of 6.5 °C km$^{-1}$. This lapse rate was selected as it is characteristic of present-day high latitude sites, e.g. Southern Chile[47], that experience climate likely comparable to the TAM prior to widespread glacier development[19]. Though the approach adopted here presumes a relationship between MSATs and ELAs (with associated uncertainties), this is partly a pragmatic simplification since ELAs are also known to vary in response to other factors including topography, aspect, and avalanching[48].

## Data availability

All data generated in this study are provided in the Supplementary Information.

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

## Acknowledgements

DEMs provided by the Byrd Polar and Climate Research Center and the Polar Geospatial Center under NSF-OPP awards 1543501, 1810976, 1542736, 1559691, 1043681, 1541332, 0753663, 1548562, 1238993, and NASA award NNX10AN61G. Computer time is provided through a Blue Waters Innovation Initiative. DEMs produced using data from DigitalGlobe, Inc. J.C.E. acknowledges support from a NERC independent fellowship award (NE/R014574/1). I.D.B., M.S., B.R.R., R.G.B., and R.P.O. acknowledge support from the Scottish Association for Environment, Geoscience and Society (SAGES) in funding R.P.O.'s Ph.D. scholarship and a sequence of annual science meetings at which the ideas for this manuscript were developed and consolidated.

## Author contributions

I.D.B., M.S., B.R.R., R.G.B., and R.P.O. conceptualised the project and methodology. I.D.B. performed all mapping. I.D.B., J.C.E., R.P., and M.D.T. performed cirque data analysis. D.J.M. analysed the climate data. I.D.B. and K.A. prepared the original draft, and all authors contributed to subsequent drafts.

## Competing interests

The authors declare no competing interests.
