## [Peer Review File · Nature Communications]

60 million years of glaciation in the Transantarctic MountainsREVIEWER COMMENTS

Reviewer #1 (Remarks to the Author):

General comments

This manuscript presents evidence suggesting that temperate warm-based glaciers were present in the Transantarctic Mountains long before the Eocene-Oligocene transition and expansion of the East Antarctic Ice Sheet: this would be a highly significant and noteworthy finding, of interest broadly to an interdisciplinary audience. The authors 1) take modern-day minimum and maximum elevations of exposed cirques to calculate equilibrium line altitudes, 2) use these elevation values to reconstruct past equilibrium line altitudes, 3) infer paleotemperature at these adjusted paleo-elevations assuming a set range of temperatures for temperate mountain glaciers, then convert it to temperature at sea level, 4) use these paleotemperature at sea level as a proxy for the date of cirque formation. Their central argument, in other words, is that cirques exposed now that were at high elevations during the Eocene-Oligocene would have been too cold for the presence of temperate mountain glaciers that formed the cirques; therefore these high elevation cirques must have formed during warmer climates, and therefore formed earlier, prior to the E-O.

Overall, the manuscript is well written, and I find it to be compelling in most respects. However, I believe that some aspects of the argument require further support and discussion. One central point is discussed in detail here, and several additional points are addressed below in the line-by-line comments:

The crux of the argument, or at least one of them, is that modern day ELA can be used to determine past ELA using paleotopographic reconstructions. I believe the passing mention in the main text (Line 72), and the discussion in Lines 238-242, do not sufficiently address the extent to which this elevation adjustment is key to the results of the analysis.

The conversion of modern day elevation to Eocene-Oligocene elevation (in the range of -1 km ~ + 2km, with a mean difference of 0.4 km) presents one of the most significant uncertainties in the analysis. The uncertainty lies in the fact that not only is the magnitude of topographic change as reconstructed by Paxman et al. (2019) a rough estimate, the spatial pattern of topographic change also is uncertain. This may be, as the authors note, the best reconstruction that we have, and it may be a more likely representative version of topography than present-day topography, but it must be clearly addressed the extent to which the key results of the manuscript depend on this reconstruction. The authors should consider demonstrating the affect of using different paleotopographic reconstructions (e.g. the Paxman paper reports median values for some inputs into the reconstruction -are there maximum/minimum scenarios that can be used?), e.g. by including different versions of Fig. 1b under these varying scenarios.

How would the reconstructed SL ELA distribution be affected by use of a slightly earlier paleotopography time slice? This would be relevant to the extent that the manuscript discusses cirques interpreted to have formed in the Oligocene and Miocene.

Furthermore, the use of Eocene-Oligocene paleotopographic reconstruction to restore modern-day ELA to paleo-ELA also implies that erosion and/or isostatic adjustment occurred between the Eocene and now to have altered the topography. To what extent are the elevation adjustments at the mapped cirques due to erosion or isostatic or other adjustments? This is important to address, because without further discussion of the inputs into the Paxman et al. reconstruction, the reader might naturally wonder: If a significant portion of the topographic adjustment is due to restoration of erosion, does this not contradict the assumption that older cirques have not been modified since their initial formation by temperate mountain glaciers? ... While Paxman et al. state that certain “relict” areas were excluded from erosional restoration, including in the TAM, it is unclear to what extent the cirques identified in this manuscript overlap with these relict areas. If erosional restoration did factor into the reconstructed areas relevant to this study, to what extent do the large uncertainties in the inputs affect the results of this study (e.g. Lindeque et al., 2016 -*Geochem, Geophys, Geosys* - <http://dx.doi.org/10.1002/2016GC006401>), if at all?

Additional line-by-line comments/concerns:

Lines 48-52

Discussions of the record of erosion in Antarctica, particularly the limitations of distal indirect proxies, raises the question of whether any direct evidence of glacial incision from the bedrock itself (i.e. the exposed cirques) were considered. Perhaps recent papers that have compiled large bodies of thermochronology data may be relevant (Sanchez et al 2021 (*Geochem, Geophys, Geosys*).

[doi/10.1029/2021GC010154](https://doi.org/10.1029/2021GC010154)

Line 72

“Once modified to topography at the EO-boundary”... - this brief sentence warrants further discussion in the main text - particularly the caveats and uncertainties of the paleotopography reconstruction used. See general comments above.

Line 73-75

“Assuming that the mean summertime air temperature...” This assumption is key to the whole argument, and here it simply refers to the Methods. Methods then refers to Supp Table. 1, which

presents evidence that temperature glaciers (but not only cirques) occupy a consistent and narrow temperature range. Is the implicit assumption then that ALL cirques (now and in the past) can only be incised/created within this temperature range? If so, this should be expressly stated, discussed, and supported with relevant citations in the main text, either here or at Line 63 ("... and thereby reconstruct paleotemperature") because it is an essential part of the argument.

Lines 127-128

The authors should address the extent to which this finding is or isn't in line with other available evidence regarding the timing of temperate glacial incision in the TAM: e.g. that lower elevation glaciers like the Beardmore in the TAM were significantly erosive and likely warm-based in the late Eocene (He et al., 2021, EPSL, <https://doi.org/10.1016/j.epsl.2021.117009>)- in that paper there are also additional data from higher elevation bedrock samples from the polar rock repository (some >40 Ma) reported in the supplement.

Lines 142-143

How are cirques in the range of ~5-10 degrees C (SL MSAT) uniquely attributed to Mid-Miocene Climatic Optimum at 15 Ma, if the entire range of 4-12 degrees C (SL MSAT) can be attributed the Oligocene (34-23 Ma)?

Lines 157-159

is it possible to cross reference the implied age of any cirques with any existing thermochronology data? Would the limited amount of erosion that occurs at a cirque be reflected in low-temperature thermochronometers like apatite U-Th/He, He 4/3 dating, or cosmogenic nuclides? It would likely be outside the scope of this manuscript to obtain new thermochronology data that directly dates the timing of erosion at high altitudes cirques in the TAM; nevertheless, it should address the relevance (if any) of existing thermochronology datasets from the literature to their key finding.

Supp Fig. 1

This figure should include the distribution of modern-day ELA elevations, prior to paleotopographic reconstruction.

Summary

My assessment is that the manuscript would be worthy of publication with moderate revisions, if the above concerns were appropriately addressed. It is well-written, and the results are potentially quite

significant. It could be improved by 1) directly addressing the major uncertainties/caveats/assumption involved in each step of its analysis, particularly that of the paleoelevation reconstruction, 2) by placing its findings in context of the literature that have sought to directly date the timing of glacier incision in the TAM, and 3) more explicitly justifying the assumptions behind the paleotemperature reconstruction

Reviewer #2 (Remarks to the Author):

Review of Barr et al. 'At least 60 million years of glaciation in the Transantarctic Mountains'

The authors combine remote sensing, glaciologic principles, and existing estimates of long-term temperature to present a comprehensive treatment of glacial conditions in the Transantarctic Mountains and constraint for the onset of alpine glaciation. Overall, the manuscript is highly ambitious and certainly provides food for thought. If many of these ideas can actually be verified (e.g., through empirical, ground-based geologic investigation), it will constitute an important advance in our understanding of glaciation in Antarctica, which, as the authors point out, is a difficult task to undertake due to extensive ice cover and the sheer length of glacial occupation. Their approach is certainly novel and draws in part from sound glacial/glacial-geomorphologic laws, namely the conditions under which cirques form and generalized relationships between cirque forms and temperature. I am concerned, however, that the manuscript paints with too broad a brush, and in so doing downplays several sources of considerable uncertainty. As written, the paper delivers temperature and age estimates of a precision I find hard to swallow considering the scale of Antarctica, the diversity of climatologic and glaciologic conditions there, and the inherent uncertainty of both the glacier-derived temperature estimates and the biological proxies to which they are compared. I am concerned that we cannot actually be so certain of many aspects used in this study, as I shall attempt to outline below. I stress that, ultimately, this is an exciting study and I look forward to seeing it in the literature, though with a considerably greater degree of transparency regarding uncertainty.

MSAT estimates: The paper assumes that the temperature at the ELA is essentially fixed, and a mean value of $3.6 \pm 2.5^\circ\text{C}$ is provided. I appreciate that this is an average, and that the specific values used to generate that average are provided in the supplementary table, but using such a specific value obscures the fact that temperate glaciers today actually occupy a staggeringly broad sweep of climatic regimes. The values in the table range from $>8^\circ\text{C}$ to $<-3^\circ\text{C}$; there are no high-latitude glaciers included; nor does this dataset include any tropical glaciers (excepting the former Chacaltaya). My point is that this $>11^\circ\text{C}$ range of MSAT at the ELA probably should not be distilled down to a tidy little value of $3.6 \pm 2.5^\circ\text{C}$, especially when temperate glaciers in Antarctica are likely to have occupied some pretty dry conditions. My attention was drawn to some modern alpine glaciers in arid tropical settings, where the ELA actually exists several hundred meters above the zero-degree isotherm (reflecting aridity). If this was the case in early Antarctica, and I question whether the authors can rule it out, then the sea-level MSATs given in

Figure 1 could be significantly inaccurate. One way round this is to be more open about the magnitude of uncertainty associated with MSAT and not try to make it appear a well-constrained value.

Along a similar vein, the authors will be well aware that many modern glaciers exhibit starkly different ELAs even among closely neighboring systems. This probably reflects the impacts of topography/hypsometry, aspect, avalanching, etc. Though I doubt such effects will shift the author's first-order interpretation of the reconstructed temperatures, it should at least be acknowledged, perhaps even reflected in the uncertainty of temperature reconstructions. In short, I am not certain the relationship between ELA and MSAT is as straightforward as presented here.

General temperature reconstructions: In 'The Onset of Glaciation', temperature estimates are provided for periods such as the early and middle Eocene. What is not clear to me, however, is to where these warm month averages apply – are we talking coastal regions? Mountaintops? The continental interior? Unless I am missing something key, this appears to be an entire continent summed up in one quick temperature value, the viability of which (i.e., what are these estimates based on) is not discussed in the text. Given that temperature estimates for the TAM during those periods are probably highly approximate, it would help to present a more realistic view of conditions and our current understanding of them.

Shift to Ice Sheet Glaciation: Lines 124-125: Similarly, these estimates of palaeo-precipitation come from published sources, but how applicable are they to the TAM? Or are we talking real generalizations? This is important, because I doubt any modern glaciologist would employ a continental average, or coastal values, in their discussions of high-altitude ELAs and glacial dynamics. I am not intending to be picky here, I just feel it is rather a generalization that should at least be delivered as such. After all, when dealing with such long timeframes, everything is going to be highly generalized and speculative.

The Return of Temperate Mountain Glaciers: Within the resolution of the various datasets used, this course of events is plausible for the TAM during the middle Miocene. However, the authors do not take into consideration the recent demonstration of cold-based glacial conditions in the central TAM (Balter-Kennedy et al., 2020) ~15 Ma, which certainly places constraint on any calls for temperate glaciation at that time. Certainly, the upper Shackleton Glacier is distant from, e.g., the McMurdo Dry Valleys and presumably experiences different climate conditions. But Balter-Kennedy documented cold-based glaciation beneath an East Antarctic Ice Sheet outlet glacier, which is presumably a lot thicker than a local cirque glacier and likelier to exhibit warm-based conditions due to overburden and geothermal trapping, etc. I find it hard to believe that temperate cirque glaciation coexisted with cold-based ice in the TAM and suspect readers will, too. I am also surprised that publication wasn't cited in relation to the Mid Miocene, since it is the sole paper to date that actually provides age control for that period. Instead, the authors cite, among others, a 1996 paper concerning the now-debunked hypothesis that fossilized wood corresponds to the Pliocene; I don't see how that study fits their suggestion that the Mid Miocene was warm.

Temperature Estimates for Different Time Periods: In my view, this is a logical course of enquiry and perhaps the only route to estimating long-term past temperature in Antarctica. But I urge the authors to remember that it is (a) biologically based and (b) a global average, which together will result in a considerable degree of uncertainty when used to establish glaciation thresholds in the high-elevation Antarctic. Curve C in Fig. 1 should at least exhibit the scale of this uncertainty, because I doubt that conditions in these polar cirques mirrored the global average so neatly.

In summary, I reiterate that I find this study intriguing, exciting, and full of potential once the sweeping generalizations that run throughout the narrative are addressed. Will this require a considerably longer paper to achieve? Probably, but it will be worth it.

REVIEWER COMMENTS

Note:

Reviewer comments are shown in blue and underlined.

Our responses are in **green and bold**.

Comments that require no response are shown in black.

Note: for clarity, we have numbered some of the reviewer comments.

Reviewer #1 (Remarks to the Author):

General comments

This manuscript presents evidence suggesting that temperate warm-based glaciers were present in the Transantarctic Mountains long before the Eocene-Oligocene transition and expansion of the East Antarctic Ice Sheet: this would be a highly significant and noteworthy finding, of interest broadly to an interdisciplinary audience. The authors 1) take modern-day minimum and maximum elevations of exposed cirques to calculate equilibrium line altitudes, 2) use these elevation values to reconstruct past equilibrium line altitudes, 3) infer paleotemperature at these adjusted paleo-elevations assuming a set range of temperatures for temperate mountain glaciers, then convert it to temperature at sea level, 4) use these paleotemperature at sea level as a proxy for the date of cirque formation. Their central argument, in other words, is that cirques exposed now that were at high elevations during the Eocene-Oligocene would have been too cold for the presence of temperate mountain glaciers that formed the cirques; therefore these high elevation cirques must have formed during warmer climates, and therefore formed earlier, prior to the E-O.

Overall, the manuscript is well written, and I find it to be compelling in most respects. However, I believe that some aspects of the argument require further support and discussion. One central point is discussed in detail here, and several additional points are addressed below in the line-by-line comments:

1.1. The crux of the argument, or at least one of them, is that modern day ELA can be used to determine past ELA using paleotopographic reconstructions. I believe the passing mention in the main text (Line 72), and the discussion in Lines 238-242, do not sufficiently address the extent to which this elevation adjustment is key to the results of the analysis. **To address this issue (and associated points raised by both reviewers), we have now added a new section (titled 'Sensitivity to input topography') to the Supplementary material where, rather than basing ELA adjustments ('modifications') on a single reconstruction of palaeotopography (i.e. the 'median' 34 Ma reconstruction from Paxman et al., 2019), we run the analysis based on nine different reconstructions (i.e. 'minimum', 'median' and 'maximum' reconstructions of topography at 34 Ma, 23 Ma, and 14 Ma - from Paxman et al., 2019). The differences between values (and associated implications for study findings) are discussed in this new section. This comparison reveals that choice of topography has some impact on how cirques are classified (whether glacier-free, temperate, or cold-based) at different time periods, but with little impact on the overall study findings.**

In the manuscript, we continue to base our results on the 'median' topography from the EO-boundary but in the methods section direct readers to the new Supplementary information to view the impacts of selecting each of the different topography reconstructions.

To emphasise some of the uncertainty introduced by different reconstructions of topography, we have 'toned down' some of the wording (e.g. by adding word such as 'likely' and 'suggest'), and have removed mention of Cretaceous (pre 66 Ma) glaciation from the text and abstract (since our findings don't really focus on this period). For this reason, we have also modified the manuscript title from:

"At least 60 million years of glaciation in the Transantarctic Mountains"

To:

"60 million years of glaciation in the Transantarctic Mountains"

Finally, we no longer refer to 'EO-boundary topography' or 'EO-ELAs' but instead refer to 'palaeo topography' and 'modified ELAs'.

1.2. The conversion of modern day elevation to Eocene-Oligocene elevation (in the range of -1 km ~ + 2km, with a mean difference of 0.4 km) presents one of the most significant uncertainties in the analysis. The uncertainty lies in the fact that not only is the magnitude of topographic change as reconstructed by Paxman et al. (2019) a rough estimate, the spatial pattern of topographic change also is uncertain. This may be, as the authors note, the best reconstruction that we have, and it may be a more likely representative version of topography than present-day topography, but it must be clearly addressed the extent to which the key results of the manuscript depend on this reconstruction. The authors should consider demonstrating the affect of using different paleotopographic reconstructions (e.g. the Paxman paper reports median values for some inputs into the reconstruction -are there maximum/minimum scenarios that can be used?), e.g. by including different versions of Fig. 1b under these varying scenarios. How would the reconstructed SL ELA distribution be affected by use of a slightly earlier paleotopography time slice? This would be relevant to the extent that the manuscript discusses cirques interpreted to have formed in the Oligocene and Miocene.

We address this in our response to comment 1.1 (above).

1.3. Furthermore, the use of Eocene-Oligocene paleotopographic reconstruction to restore modern-day ELA to paleo-ELA also implies that erosion and/or isostatic adjustment occurred between the Eocene and now to have altered the topography. To what extent are the elevation adjustments at the mapped cirques due to erosion or isostatic or other adjustments? This is important to address, because without further discussion of the inputs into the Paxman et al. reconstruction, the reader might naturally wonder: If a significant portion of the topographic adjustment is due to restoration of erosion, does this not contradict the assumption that older cirques have not been modified since their initial formation by temperate mountain glaciers? ... While Paxman et al. state that certain "relict" areas were excluded from erosional restoration, including in the TAM, it is unclear to what extent the cirques identified in this manuscript overlap with these relict areas. If erosional restoration did nfactor into the reconstructed areas relevant to this study, to what extent do the large uncertainties in the inputs affect the results of this study (e.g. Lindeque et al., 2016 -Geochem, Geophys, Geosys - <http://dx.doi.org/10.1002/2016GC006401>), if at all?

Uncertainties introduced by reconstructions of palaeotopography are now exemplified in the Supplementary information, as outlined in our response to comment 1.1 (above).

It is not possible to address the extent to which elevation adjustments at the mapped cirques are due to erosion or isostatic or other adjustments. However, in newly added

figures (supplementary Fig 1c and supplementary Fig 3b) we now report the elevation differences between 'modified' and 'unmodified' ELAs. These data illustrate that 'modified' ELAs are typically higher than 'unmodified' ELAs (though this varies from cirque-to-cirque). In the reconstructions developed by Paxman et al (2019) the scale over which topographic adjustments (relative to present) are made (i.e. palaeotopography grids are produced at 5 km horizontal resolution and smoothed with a 10 km Gaussian filter) means that erosional restoration within cirques (smaller features than the grid resolution) is unlikely to be relevant. A small number (<5%) of cirques analysed in this study lie within the 'relict' areas defined by Paxman et al (2019). However, beyond visual comparison with Fig. 1 in Paxman et al (2019) it is not possible to be more precise/definitive than this, and no associated modification has been made to the manuscript.

Additional line-by-line comments/concerns:

Lines 48-52

Discussions of the record of erosion in Antarctica, particularly the limitations of distal indirect proxies, raises the question of whether any direct evidence of glacial incision from the bedrock itself (i.e. the exposed cirques) were considered. Perhaps recent papers that have compiled large bodies of thermochronology data may be relevant (Sanchez et al 2021 (Geochem, Geophys, Geosys). doi/10.1029/2021GC010154

In the section titled 'The onset of glaciation' we now make a direct comparison with some of the thermochronologic literature, and have added the following text:

"In addition, thermochronologic studies find evidence of enhanced exhumation in the TAM during the Late Eocene and Early Oligocene (see below), presumed to reflect increased erosion by temperate glaciers^{25,26}."

Line 72

"Once modified to topography at the EO-boundary"... - this brief sentence warrants further discussion in the main text - particularly the caveats and uncertainties of the paleotopography reconstruction used. See general comments above.

We address this in our response to comment 1.1 (above).

Line 73-75

"Assuming that the mean summertime air temperature..." This assumption is key to the whole argument, and here it simply refers to the Methods. Methods then refers to Supp Table. 1, which presents evidence that temperature glaciers (but not only cirques) occupy a consistent and narrow temperature range. Is the implicit assumption then that ALL cirques (now and in the past) can only be incised/created within this temperature range? If so, this should be expressly stated, discussed, and supported with relevant citations in the main text, either here or at Line 63 ("... and thereby reconstruct paleotemperature") because it is an essential part of the argument.

To address this, we have now re-worded this section of the text, from:

"In this study we analyse the altitudes of glacier-free cirques across the Transantarctic Mountains (TAM), use this information to estimate past glacier ELAs, and thereby reconstruct palaeotemperatures."

To:

“Assuming that these cirques developed when occupied by temperate glaciers (see Methods)²⁰ and using modern temperate glaciers (and the climatic conditions they experience) as analogues¹¹, we reconstruct palaeotemperatures from these ELAs.”

we have also added the following to the ‘Palaeotemperature reconstruction’ section of the methods:

“Because of the requirement for efficient and focused subglacial erosion (driven by localised sliding and the flushing of subglacial debris), it is generally assumed that cirques primarily develop (i.e. initiate and grow) when they are occupied by comparatively small temperate (i.e. warm-based) mountain glaciers^{20,39}. Thus, we presume that the climatic conditions which facilitate present-day temperate glaciation in various regions globally also prevailed in the TAM when the ice-free cirques analysed in this study were being formed¹¹.”

Lines 127-128

The authors should address the extent to which this finding is or isn't in line with other available evidence regarding the timing of temperate glacial incision in the TAM: e.g. that lower elevation glaciers like the Beardmore in the TAM were significantly erosive and likely warm-based in the late Eocene (He et al., 2021, EPSL, <https://doi.org/10.1016/j.epsl.2021.117009>)- in that paper there are also additional data from higher elevation bedrock samples from the polar rock repository (some >40 Ma) reported in the supplement.

As noted above, in order to compare findings with some of the thermochronologic literature (including the study mentioned by the reviewer) we have now added the following text (and relevant citations) to the manuscript:

“In addition, thermochronologic studies find evidence of enhanced exhumation in the TAM during the Late Eocene and Early Oligocene (see below), presumed to reflect increased erosion by temperate glaciers^{25,26}.”

Lines 142-143

How are cirques in the range of ~5-10 degrees C (SL MSAT) uniquely attributed to Mid-Miocene Climatic Optimum at 15 Ma, if the entire range of 4-12 degrees C (SL MSAT) can be attributed the Oligocene (34-23 Ma)?

Cirques in the range of ~5-10°C are not uniquely attributed to a single period but are presumed to have been occupied by temperate glaciers during both the Oligocene (34-23 Ma) and Mid-Miocene Climatic Optimum (15 Ma).

Note: for reasons outlined in response to reviewer comment 2.3 (below), the MSAT range used for the Mid-Miocene Climatic Optimum is now $7 \pm 4^\circ\text{C}$ rather than ~5-10°C.

Lines 157-159

is it possible to cross reference the implied age of any cirques with any existing thermochronology data? Would the limited amount of erosion that occurs at a cirque be reflected in low-temperature thermochronometers like apatite U-Th/He, He 4/3 dating, or cosmogenic nuclides? It would likely be outside the scope of this manuscript to obtain new thermochronology data that directly dates the timing of erosion at high altitudes cirques in the TAM; nevertheless, it should address the relevance (if any) of existing thermochronology datasets from the literature to their key finding.

As noted in our response to previous comments (above) we have now made reference to thermochronologic studies in the section titled ‘The onset of glaciation’.

Supp Fig. 1

This figure should include the distribution of modern-day ELA elevations, prior to paleotopographic reconstruction.

This figure has now been modified to include two additional panels: panel (a) showing the distribution of ‘unmodified’ ELAs (what the reviewer refers to as ‘modern-day’ ELAs), and panel (c) showing the elevation difference between ‘modified’ and ‘unmodified’ ELAs (partly in response to comment 1.3, above).

Summary

My assessment is that the manuscript would be worthy of publication with moderate revisions, if the above concerns were appropriately addressed. It is well-written, and the results are potentially quite significant. It could be improved by 1) directly addressing the major uncertainties/caveats/assumption involved in each step of its analysis, particularly that of the paleoelevation reconstruction, 2) by placing its findings in context of the literature that have sought to directly date the timing of glacier incision in the TAM, and 3) more explicitly justifying the assumptions behind the paleotemperature reconstruction.

We thank the reviewer, and hope that we have now addressed each of these concerns through modifications to the manuscript and associated material (as outlined in our responses above).

Reviewer #2 (Remarks to the Author):

Review of Barr et al. ‘At least 60 million years of glaciation in the Transantarctic Mountains’

The authors combine remote sensing, glaciologic principles, and existing estimates of long-term temperature to present a comprehensive treatment of glacial conditions in the Transantarctic Mountains and constraint for the onset of alpine glaciation. Overall, the manuscript is highly ambitious and certainly provides food for thought. If many of these ideas can actually be verified (e.g., through empirical, ground-based geologic investigation), it will constitute an important advance in our understanding of glaciation in Antarctica, which, as the authors point out, is a difficult task to undertake due to extensive ice cover and the sheer length of glacial occupation. Their approach is certainly novel and draws in part from sound glacial/glacial-geomorphologic laws, namely the conditions under which cirques form and generalized relationships between cirque forms and temperature. I am concerned, however, that the manuscript paints with too broad a brush, and in so doing downplays several sources of considerable uncertainty. As written, the paper delivers temperature and age estimates of a precision I find hard to swallow considering the scale of Antarctica, the diversity of climatologic and glaciologic conditions there, and the inherent uncertainty of both the glacier-derived temperature estimates and the biological proxies to which they are compared. I am concerned that we cannot actually be so certain of many aspects used in this study, as I shall attempt to outline below. I stress that, ultimately, this is an exciting study and I look forward to seeing it in the literature, though with a considerably greater degree of transparency regarding uncertainty.

2.1. MSAT estimates: The paper assumes that the temperature at the ELA is essentially fixed, and a mean value of \$3.6 \pm 2.5^{\circ}\text{C}\$ is provided. I appreciate that this is an average, and that the specific values used to generate that average are provided in the supplementary table, but using such a specific value obscures the fact that temperate glaciers today actually occupy a staggeringly broad sweep of climatic regimes. The values in the table range from \$>8^{\circ}\text{C}\$ to \$<-3^{\circ}\text{C}\$; there are no high-latitude glaciers included; nor does this dataset include any tropical glaciers (excepting the former Chacaltaya). My point is that this \$>11^{\circ}\text{C}\$ range of MSAT at the ELA probably should not be distilled down to a tidy little value of \$3.6 \pm 2.5^{\circ}\text{C}\$, especially when temperate glaciers in Antarctica are likely to have occupied some pretty dry conditions. My attention was drawn to some modern alpine glaciers in arid tropical settings, where the ELA actually exists several hundred meters above the zero-degree isotherm (reflecting aridity). If this was the case in early Antarctica, and I

question whether the authors can rule it out, then the sea-level MSATs given in Figure 1 could be significantly inaccurate. One way round this is to be more open about the magnitude of uncertainty associated with MSAT and not try to make it appear a well-constrained value.

In the present study, we have used the mean $\pm 1\sigma$ of MSAT at modern temperate glaciers to reflect palaeo conditions following Rose et al (2013). We accept that this ‘simplifies’ the broad range of MSAT that modern temperate glaciers experience, but consider the mean $\pm 1\sigma$ to usefully (i.e. in a way that can be applied to palaeo glaciers) reflect much of the variability in the data.

Though we continue to use this $3.6 \pm 2.5^\circ\text{C}$ value in the manuscript, we have now added the following text to the ‘Palaeotemperature reconstruction’ of the methods section:

“This value represents the mean $\pm 1\sigma$, but it is worth noting that the present-day MSAT values for these temperate glaciers (Supplementary Table 1) range from -3.7°C to 8.8°C . Though this 12.4°C range is considerable, the 10th to 90th percentile range is only 5.9°C (i.e. 0.9°C to 6.8°C) and very similar to the mean $\pm 1\sigma$ range. For this reason, and following previous studies of Antarctic cirques¹¹, in this study, we assume that this mean $\pm 1\sigma$ MSAT also applied at the ELAs of former temperate mountain glaciers in Antarctica¹¹.”

2.2. Along a similar vein, the authors will be well aware that many modern glaciers exhibit starkly different ELAs even among closely neighboring systems. This probably reflects the impacts of topography/hypsometry, aspect, avalanching, etc. Though I doubt such effects will shift the author’s first-order interpretation of the reconstructed temperatures, it should at least be acknowledged, perhaps even reflected in the uncertainty of temperature reconstructions. In short, I am not certain the relationship between ELA and MSAT is as straightforward as presented here.

This is partly addressed in our response to comment 2.1 (above). We have also added the following text to the Methods section:

“Though the approach adopted here presumes a relationship between MSATs and ELAs (with associated uncertainties), this is partly a pragmatic simplification since ELAs are also known to vary in response to other factors including topography, aspect, and avalanching⁵⁰.”

2.3. General temperature reconstructions: In ‘The Onset of Glaciation’, temperature estimates are provided for periods such as the early and middle Eocene. What is not clear to me, however, is to where these warm month averages apply – are we talking coastal regions? Mountaintops? The continental interior? Unless I am missing something key, this appears to be an entire continent summed up in one quick temperature value, the viability of which (i.e., what are these estimates based on) is not discussed in the text. Given that temperature estimates for the TAM during those periods are probably highly approximate, it would help to present a more realistic view of conditions and our current understanding of them.

To address some of the uncertainty about the palaeotemperature estimates, we have added a new supplementary table (Supplementary Table 2), which provides details of these data. Readers are now directed to this table in the main text, the methods, and in the caption for Fig. 1c.

This table indicates that the palaeo MSAT estimates used in this study are coastal (often from the Ross Sea, adjacent to the Transantarctic Mountains) and reflect conditions at sea level. The one exception in the original manuscript was the $\sim 5^\circ\text{C}$ Mid-Miocene Climatic optimum (MMCO) MSAT estimate from Francis & Hill (1996). To avoid this situation (and in response to reviewer point 2.5 below), we no longer use this temperature estimate, and remove the associated citation from the manuscript.

Instead, we use sea level MSAT data from Feakins et al. (2012) (i.e. MSAT = $7 \pm 4^\circ\text{C}$), supported by Warny et al. (2019). To emphasise that these palaeotemperature estimates reflect conditions at sea level, in the manuscript we now refer to them as “*published estimates of SL palaeo MSAT*”.

2.4. Shift to Ice Sheet Glaciation: Lines 124-125: Similarly, these estimates of palaeoprecipitation come from published sources, but how applicable are they to the TAM? Or are we talking real generalizations? This is important, because I doubt any modern glaciologist would employ a continental average, or coastal values, in their discussions of high-altitude ELAs and glacial dynamics. I am not intending to be picky here, I just feel it is rather a generalization that should at least be delivered as such. After all, when dealing with such long timeframes, everything is going to be highly generalized and speculative.

To address some of the uncertainty about the palaeoprecipitation estimates, we have added a new supplementary table (Supplementary Table 3), which provides details of these data. With one exception, these palaeoprecipitation estimates come from West Antarctica, the Ross Sea (adjacent to the TAM) and from the TAM. The exception is the 1250 mm estimate for the Late Eocene, which comes from Prydz Bay (East Antarctica), and has now been removed from the manuscript. Note: In the manuscript, palaeoprecipitation data are only used to illustrate a general decline in values towards the present and are not used for any analyses.

2.5. The Return of Temperate Mountain Glaciers: Within the resolution of the various datasets used, this course of events is plausible for the TAM during the middle Miocene. However, the authors do not take into consideration the recent demonstration of cold-based glacial conditions in the central TAM (Balter-Kennedy et al., 2020) ~15 Ma, which certainly places constraint on any calls for temperate glaciation at that time. Certainly, the upper Shackleton Glacier is distant from, e.g., the McMurdo Dry Valleys and presumably experiences different climate conditions. But Balter-Kennedy documented cold-based glaciation beneath an East Antarctic Ice Sheet outlet glacier, which is presumably a lot thicker than a local cirque glacier and likelier to exhibit warm-based conditions due to overburden and geothermal trapping, etc. I find it hard to believe that temperate cirque glaciation coexisted with cold-based ice in the TAM and suspect readers will, too. I am also surprised that publication wasn't cited in relation to the Mid Miocene, since it is the sole paper to date that actually provides age control for that period. Instead, the authors cite, among others, a 1996 paper concerning the now-debunked hypothesis that fossilized wood corresponds to the Pliocene; I don't see how that study fits their suggestion that the Mid Miocene was warm.

As noted in our response to reviewer point 2.3, we no longer use the temperature estimate from Francis and Hill (1996) and have removed the associated citation from the manuscript.

In relation to Balter-Kennedy et al. (2020), they certainly find evidence for cold-based ice (ice sheet) in the central TAM dating back to ~14.5 Ma. However, this doesn't really contradict our findings, since we suggest that temperate glaciers may last have occupied parts of the TAM ~15 Ma (i.e. during the MMCO), after which cold-based glaciation came to dominate. We have now added a reference to Balter-Kennedy et al. (2020) in the section of the manuscript that mentions the post-MMCO cooling, aridity, and associated switch to cold-based glaciation.

2.6. Temperature Estimates for Different Time Periods: In my view, this is a logical course of enquiry and perhaps the only route to estimating long-term past temperature in Antarctica. But I urge the authors to remember that it is (a) biologically based and (b) a global average, which together will result in a considerable degree of uncertainty when used to establish glaciation thresholds in the high-elevation Antarctic. Curve C in Fig. 1 should at least exhibit the scale of this uncertainty, because I doubt that conditions in these polar cirques mirrored the global average so neatly.

The temperature data in Fig. 1b (formerly 1c) are indeed global (as mentioned in the figure and caption), but are simply used to illustrate broad trends, and not used to produce any of the results in this study. As noted above, the analysis conducted in the present study is based on palaeo temperature data from Antarctica. To emphasise this, we now explicitly state in the caption for Fig. 1c that these are ‘Antarctic’ data. We have also added a new supplementary table (Supplementary Table 2), which provides some details of these Antarctic palaeotemperature data.

In summary, I reiterate that I find this study intriguing, exciting, and full of potential once the sweeping generalizations that run throughout the narrative are addressed. Will this require a considerably longer paper to achieve? Probably, but it will be worth it.

Again, we thank the reviewer, and hope that we have now addressed each of these concerns through modifications to the manuscript and associated material.

Further modifications:

In making the changes outlined above, we noticed that:

Lewis, A.R. *et al.* Mid-Miocene cooling and the extinction of tundra in continental Antarctica, *PNAS* 105, 10676–10680 (2008).

Was incorrectly cited as:

32. Lewis, A.R., Marchant, D.R., Ashworth, A.C., Hemming, S.R. & Machlus, M.L. Major middle Miocene global climate change: Evidence from East Antarctica and the Transantarctic Mountains. *Geol. Soc. Am. Bull.* 119, 1449–1461 (2007).

We have now corrected this in the reference list.

In Fig. 1, panels (b) and (c) have now been switched to match the order they are mentioned in the text. The x-axis of graphs in these panels has also been reversed, so that geological time is shown progressing from left (oldest) to right (youngest), which seems more intuitive given the manuscript structure.

In Fig. 2, pie charts for the different sub-regions have now been removed. First, to de-clutter this image. Second, because these regional differences are not discussed in the text.

In Supplementary table 1, glaciers are now listed (sorted) by their MSATs (from lowest to highest), and we now specify (in the table) that MSAT values are recorded in °C.

In compliance with Nature Communications formatting instructions we have now added a final paragraph beginning with the phrase “We show” to the introduction – i.e.

“We show that mountain glaciers were likely present in the TAM during the Late Palaeocene (~60–56 Ma) and middle Eocene (~48–40 Ma). Temperate (warm-based) glaciers were widespread during the Late Eocene (~40–34 Ma) and, in reduced numbers, during the Oligocene (~34–23 Ma). By the Early Miocene (~23–15 Ma) it is likely that all of the cirques in the TAM were submerged by ice sheets, and/or occupied by mostly cold-based glaciers, but that some temperate glaciers were present during the Miocene Climatic Optimum (~15 Ma), before a widespread switch to cold-based glaciation, which continues to the present-day.”

REVIEWERS' COMMENTS

Reviewer #1 (Remarks to the Author):

Overall, I find that the resubmission have satisfactorily addressed most of my concerns with the initial submission of the manuscript, particularly the uncertainty and assumptions associated with their argument.

I have one remaining concern related to the authors' response on one point (copied and discussed below). Besides this, it is my opinion that the findings are significant enough to warrant publication in this journal after the issue below is addressed. This paper will provide ample food for debate in the community. The use of modern ELA as a paleotemperature proxy and to implicitly date a glacier is novel, and publication of the manuscript would lay the foundation for improvements of the analysis, should there be improvements in paleotopographic reconstructions in the future.

Author's response: "It is not possible to address the extent to which elevation adjustments at the mapped cirques are due to erosion or isostatic or other adjustments. However, in newly added figures (supplementary Fig 1c and supplementary Fig 3b) we now report the elevation differences between 'modified' and 'unmodified' ELAs. These data illustrate that 'modified' ELAs are typically higher than 'unmodified' ELAs (though this varies from cirque-to-cirque). In the reconstructions developed by Paxman et al (2019) the scale over which topographic adjustments (relative to present) are made (i.e. palaeotopography grids are produced at 5 km horizontal resolution and smoothed with a 10 km Gaussian filter) means that erosional restoration within cirques (smaller features than the grid resolution) is unlikely to be relevant."

These additional supplementary figures show that a key result of the author's analysis is that, as a whole, modified ELAs of the exposed cirques would be on average higher in the Eocene than now. This means that either (end member 1) that the surface was lowered without erosion (e.g. post TAM rifting thermal subsidence; ice sheet loading) or (end member 2) surface was lowered entirely due to erosion. Even if it is not possible to address to what extent erosion was responsible for the topography change, it would be critical to discuss how this impacts their analysis. The 5km grid resolution smoothed over 10km would still imply that the topographic change, as reconstructed at any given grid point, applies over the entire 5 km grid cell. If a cirque's ELA is modified to increase by 500 m relative to the present, a critical assumption here seems to be that, to the extent that a portion of this elevation change was due to erosion, the cirque morphology must have been preserved despite that amount of erosion (i.e. erosion was spatially uniform). i.e. that modification of a cirque's elevation was not associated with

modification of its morphology (such that it is still recognizable as a cirque today). I think this assumption is plausible, given that a substantial portion of elevation change can likely be attributed to ice sheet loading/unloading (Fig. 3c of Paxman), but this should be explicitly stated and discussed in the manuscript somewhere.

Reviewer #2 (Remarks to the Author):

Having reviewed the revised manuscript, I am of the opinion that the authors have addressed the major concerns of both reviewers sufficiently, particularly with regard to transparency. There remains a measurable degree of extrapolation in the climatic interpretation of results, but I think that is overall a good thing - readers can judge the validity of those interpretations for themselves, as in all science. I am happy to recommend publication of the revised manuscript.

REVIEWER COMMENTS

Note:

Reviewer comments are shown in blue and underlined.

Our responses are in **green and bold**.

Comments that require no response are shown in black.

Reviewer #1 (Remarks to the Author):

Overall, I find that the resubmission have satisfactorily addressed most of my concerns with the initial submission of the manuscript, particularly the uncertainty and assumptions associated with their argument.

I have one remaining concern related to the authors' response on one point (copied and discussed below). Besides this, it is my opinion that the findings are significant enough to warrant publication in this journal after the issue below is addressed. This paper will provide ample food for debate in the community. The use of modern ELA as a paleotemperature proxy and to implicitly date a glacier is novel, and publication of the manuscript would lay the foundation for improvements of the analysis, should there be improvements in paleotopographic reconstructions in the future.

Author's response: "It is not possible to address the extent to which elevation adjustments at the mapped cirques are due to erosion or isostatic or other adjustments. However, in newly added figures (supplementary Fig 1c and supplementary Fig 3b) we now report the elevation differences between 'modified' and 'unmodified' ELAs. These data illustrate that 'modified' ELAs are typically higher than 'unmodified' ELAs (though this varies from cirque-to-cirque). In the reconstructions developed by Paxman et al (2019) the scale over which topographic adjustments (relative to present) are made (i.e. palaeotopography grids are produced at 5 km horizontal resolution and smoothed with a 10 km Gaussian filter) means that erosional restoration within cirques (smaller features than the grid resolution) is unlikely to be relevant."

These additional supplementary figures show that a key result of the author's analysis is that, as a whole, modified ELAs of the exposed cirques would be on average higher in the Eocene than now. This means that either (end member 1) that the surface was lowered without erosion (e.g. post TAM rifting thermal subsidence; ice sheet loading) or (end member 2) surface was lowered entirely due to erosion. Even if it is not possible to address to what extent erosion was responsible for the topography change, it would be critical to discuss how this impacts their analysis. The 5km grid resolution smoothed over 10km would still imply that the topographic change, as reconstructed at any given grid point, applies over the entire 5 km grid cell. If a cirque's ELA is modified to increase by 500 m relative to the present, a critical assumption here seems to be that, to the extent that a portion of this elevation change was due to erosion, the cirque morphology must have been preserved despite that amount of erosion (i.e. erosion was spatially uniform). i.e. that modification of a cirque's elevation was not associated with modification of its morphology (such that it is still recognizable as a cirque today). I think this assumption is plausible, given that a substantial portion of elevation change can likely be attributed to ice sheet loading/unloading (Fig. 3c of Paxman), but this should be explicitly stated and discussed in the manuscript somewhere.

In the Supplementary section titled 'Sensitivity to input topography' we have now added the following paragraph to address this issue:

“Finally, supplementary Fig 3b indicates that during the past ~34 Ma, in most cases, glacier-free cirques in the TAM were located at higher elevations than at present. However, as noted in the Methods section, it is unlikely that (in terms of size and shape) the cirques themselves were modified extensively over this period. This might indicate that a substantial portion of this elevation change was driven by ice sheet loading/unloading, rather than by cirque-focused erosion.”

Reviewer #2 (Remarks to the Author):

Having reviewed the revised manuscript, I am of the opinion that the authors have addressed the major concerns of both reviewers sufficiently, particularly with regard to transparency. There remains a measurable degree of extrapolation in the climatic interpretation of results, but I think that is overall a good thing - readers can judge the validity of those interpretations for themselves, as in all science. I am happy to recommend publication of the revised manuscript.